# Hey Max, Can You Help Me? An Intuitive Virtual Assistant for Industrial Robots

**Chen Li** [1,*], **Dimitrios Chrysostomou** [1], **Daniela Pinto** [1], **Andreas Kornmaaler Hansen** [2,3], **Simon Bøgh** [1] **and Ole Madsen** [1]

1   Robotics and Automation Group, Department of Materials and Production, Aalborg University, Fibigerstraede 16, 9220 Aalborg, Denmark
2   Center for Industrial Production, Department of Materials and Production, Aalborg University, Fibigerstraede 16, 9220 Aalborg, Denmark
3   Department of Technology and Business, University College of Northern Denmark, Sofiendalsvej 60, 9000 Aalborg, Denmark
*   Correspondence: cl@mp.aau.dk; Tel.: +45-52-645-576

**Abstract:** Assisting employees in acquiring the knowledge and skills necessary to use new services and technologies on the shop floor is critical for manufacturers to adapt to Industry 4.0 successfully. In this paper, we employ a learning, training, assistance-formats, issues, tools (LTA-FIT) approach and propose a framework for a language-enabled virtual assistant (VA) to facilitate this adaptation. In our system, the human–robot interaction is achieved through spoken natural language and a dashboard implemented as a web-based application. This type of interaction enables operators of all levels to control a collaborative robot intuitively in several industrial scenarios and use it as a complementary tool for developing their competencies. Our proposed framework has been tested with 29 users who completed various tasks while interacting with the proposed VA and industrial robots. Through three different scenarios, we evaluated the usability of the system for LTA-FIT based on an established system usability scale (SUS) and the cognitive effort required by the users based on the standardised NASA-TLX questionnaire. The qualitative and quantitative results of the study show that users of all levels found the VA user friendly with low requirements for physical and mental effort during the interaction.

**Keywords:** natural language processing; virtual assistant; human–robot interaction; usability studies; NASA TLX; system usability scale

## 1. Introduction

Digitalization is changing the manufacturing world. The exponential growth in digital technologies provides manufacturing companies with possibilities for introducing new products, processes, and services, many of which have the potential to be disruptive. Hence, the manufacturing industry has adapted to Industry 4.0 values and is slowly transitioning to the Industry 5.0 era [1]. However, due to the complexity of the manufacturing tasks and the speed of the technological development, the adoption of new technologies remains a grand challenge [2].

As the nature of many proposed solutions typically involves multidisciplinary activities, operators with years of experience and domain knowledge are needed. However, in modern industrial settings, such expertise is not always readily available, and new, more flexible approaches for knowledge acquisition and dissemination of learned experience should be utilized [3]. One revolutionary model used to assist workers in their learning and training is the learning, training, assistance-formats, issues, tools (LTA-FIT) model proposed by [4]. The model offers a flexible approach to assist the re-qualification and training of workers in new digital tools.

In this work, we expand the proof-of-concept we presented previously [5], in which we proposed a virtual assistant (VA) built on a natural language processing architecture used as a digital interpretation of the LTA-FIT model in a smart factory. The system provides additional instructions to facilitate a basic understanding of the steps of an assembly process of a mobile phone mockup, assists the workers through the necessary training process, and then answers workers' task-related questions. Because human operators are at the center of the LTA-FIT model, we performed an extensive user study to evaluate the system's usability and the users' acceptance, comfort, satisfaction, and task load. We used the proposed framework to complete an assembly task by using the eighth iteration of our autonomous industrial mobile manipulator (AIMM), named Little Helper 8 (LH8) (Figure 1). Little Helper is a series of mobile manipulators with an integrated industrial robot arm and mobile platform to offer flexible solutions to industrial settings [6]. The eighth iteration consists of a Franka Emika Panda collaborative robot mounted on a MiR200 omnidirectional mobile platform.

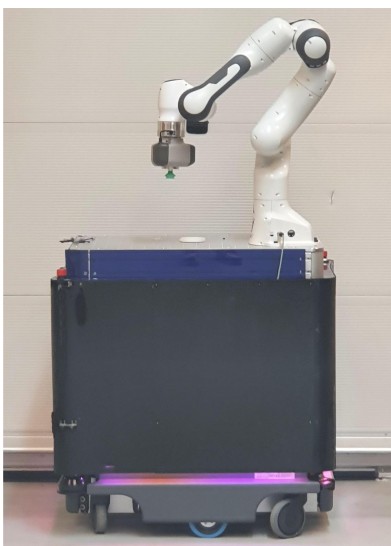

**Figure 1.** The autonomous industrial mobile manipulator Little Helper, in which our proposed virtual assistant is integrated.

*Contribution*

In this study, we design and create a language-enabled VA for human–robot interaction in LTA-FIT of manufacturing tasks that enhance the user experience while maintaining a high level of system usability. Our specific contributions are:

- a novel VA trained on the state-of-the-art bidirectional encoder representations from transformers (BERT) [7] model with a high human intent-recognition capability focusing on the industrial robots and satisfies downstream assistance needs of factories for employees to successfully obtain the acquired knowledge and skills for manufacturing tasks;
- three common manufacturing scenarios are selected and analyzed in a genuine production setting with an emphasis on each level of LTA-FIT. To our knowledge, this is the first time an industrial-oriented VA has been integrated into an industrial setting with the goal of providing LTA-FIT for shop floor workers; and
- a user study of human–machine interaction with both the VA and the robot. This analysis evaluates the system's usability, the perceived comfort, and mental load while working on the task, and the comfortability, perceived competence, performance expectancy, and effort expectancy.

In the rest of the paper, we present an overview of related literature in Section 2, the architecture of the proposed framework in Section 3 and the selected LTA-FIT scenarios in

Section 4. Sections 5 and 6 present the experimental setup and analyze the obtained results while we reflect on the results and conclude the paper in Section 7.

## 2. Related Work

As we discussed before, the transition to the so-called Industry 5.0 era requires specialists and domain experts to be able to learn new concepts quickly and improve their qualifications intuitively [8]. A general term used often for such workplace training activities is work-based or work-integrated learning [9]. It is related to all learning generated directly through workplace considerations and concerns. It is typically an unplanned and informal process and either appears organically through regular development and problem-solving at work or as part of workplace training or coaching in response to identified work-related issues [10]. It has been proven to be more effective compared to traditional classroom teaching, mainly when it tackles real work issues, and it can improve professional work readiness, self-efficacy, and team skills [11].

In the context of industrial automation and manufacturing processes, the learning principles of work-based learning have been incorporated effectively into the concept of learning factories [12,13]. They constitute a paradigm of close collaboration between industry parties and universities to offer a flexible industrial environment where the involved stakeholders can introduce and test artificial intelligence (AI) techniques [14], swarm production concepts [15], and virtual recommissioning methods [16].

However, although learning factories can support proof-of-concept integration, they do not scale up adequately to real industrial challenges, as human operators, who should work in close collaboration with various machines or robots, lack the required qualifications [17]. Therefore, several employee qualification and competency-building models have been developed whereby a variety of personal, social, action-related, and domain-related competencies can be improved [18,19].

In our work in particular, where human–robot collaboration is the focus, we employ the LTA-FIT qualification model, and based on the lessons learned from [20] we propose a framework for a language-enabled virtual assistant (VA)-integrated system with the latest iteration of our AIMM [21] and deployed it in our own learning factory [22].

In recent years, voice-enabled interaction and VAs, in particular, have become an integral element in the interaction of humans with machines in various domains [23]. In the area of teaching and learning, Prgorskly et al. [24] proposed a virtual learning assistant as an assessment tool by which to support online learners' self-regulation in online learning, whereas Preece et al. [25] investigated the effectiveness of a controlled natural language-based conversational interface for human–machine interaction regarding locally observed collective knowledge. In the manufacturing domain, Barbosa et al. [26] created a VA, which supported the training process of the technical personnel in industrial plant operators, Casillo et al. [27] built a chatbot for the training of employees in an Industry 4.0 scenario, and Zimmer et al. [28] developed a VA to assist the ramp-up process of an assembly system. In our previous work, we have presented a VA named Bot-X to assist the workers in handling a variety of complex manufacturing tasks, e.g., order processing, and production execution [29]. Naturally, VAs, dialogue systems, and conversational agents have also facilitated scenarios wherein humans work closely with robots. Maksymova et al. [30] used several models for voice control of an industrial robot in the context of a collaborative assembly whereas Bingol and Aydogmus [31] evaluated deep neural networks for the classification of commands in a natural speech-recognition system for a collaborative robot task.

Despite the vast amount of research that has been performed in all these domains, VAs are still not readily accepted as the preferred means of interaction with machines or robots. Many researchers have concluded that several challenges remain to be resolved. Russo et al. [32] focused their research on using dialogue systems and conversational agents for patients with dementia, where they concluded that the delay of the robot in responding to patients increased the cognitive load of the patients. Similarly, Bernard [33] and

Rawassizadeh et al. [23] manifest that the use of VAs increases the cognitive requirements of the users. Additionally, Skantze [34] evaluated conversational agents in human–robot interaction (HRI) and concluded that natural timing in conversation systems is one of the essential factors for fluent human–human interaction, but it is also a challenging feature to adopt in VAs. He also emphasized that broken words and unclear sentences often hinder the performance of the VA and consequently increase the discomfort of the users.

Therefore, in this paper, we have a strong interest in evaluating our VA by human operators in an industrial scenario. We have enhanced our VA with natural timing capabilities, and we performed a user study focused on tracking the comfortability of the users and the usability and cognitive requirements of our overall system. Based on the learnings from other researchers [35–37], and our lessons learned from standardising HRI and human–robot collaboration (HRC) experiments before [38,39], we evaluated the system's usability with the system usability scale (SUS) [40], the perceived mental load while working on the task, based on the NASA task load index (TLX) [41], the perceived competence, performance expectancy, and effort expectancy metrics inspired by [35], and the comfortability scale based on our uniquely designed questionnaire.

### 3. System Overview

This section describes the architecture of the proposed natural language-enabled VA, named Max. A preliminary development step is presented in our previous work [21]. Following the client-server (CS) style architecture, language and robot services are hosted on the Max server side. The Max client is composed of Microsoft cognitive speech service and the robot control module. RESTful API requests support the communication between the Max server and Max client. The Max client provides a voice interface (i.e., microphone and speaker) to interact with the human operator and leverages the RESTful APIs to communicate with the shop floor robots. The overall architecture is showcased in Figure 2 while a description of each component is presented in the remainder of the section.

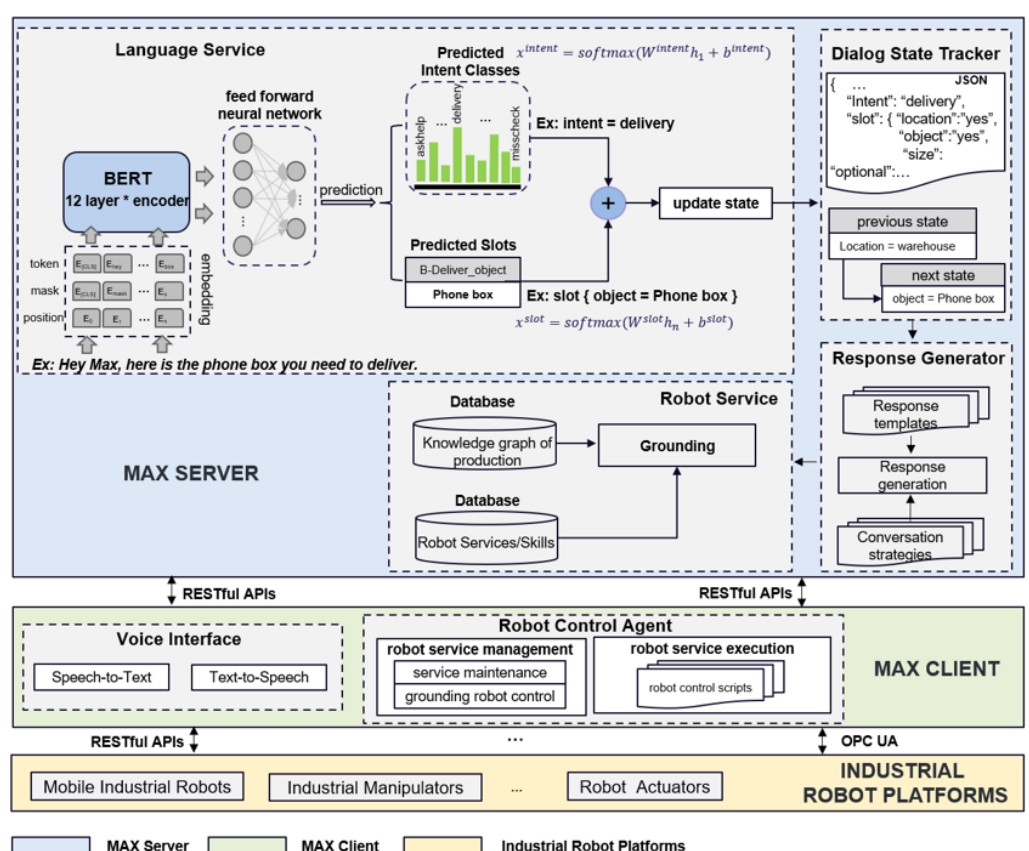

**Figure 2.** The architecture of the proposed natural language-enabled VA.

### 3.1. Language Services

**Intent Recognition**. Two services are provided for human-intent recognition—rule-based keyword extraction and neural network-based intent recognition. Due to its simplicity and quick response time, learning and training scenarios rely heavily on the keyword-extraction approach. Max, on the other hand, should be able to precisely grasp the operator's intent and directions in order to assist with everyday logistic activities in the assistance scenario.

We fine-tuned the pretrained basic BERT model (see language service part of Figure 3), which provides contextualized sentence representation and can learn the meaning of the words in the given context to predict the operator's intent and key slots from the operator's utterance. The model has 12 layers, 768 hidden units, 12 attention heads, and 110 M parameters fine-tuned with a feed-forward neural network (FFNN) layer and Softmax as the activation function to normalize the output of the model. A series of sequential operations (masking (15% of the words), tokenisation, and embeddings) are applied to the user's utterance to generate the input representation. A special symbol, [CLS], is added as the first token to indicate the beginning of the user's utterance. The embedded utterances are passed through 12 transformer encoders. The last FFNN takes the results of the BERT model as input and outputs a possible user's intent ($X^{intent}$) and slots ($X^{slot}$). The model is trained and validated on a dialogue dataset containing the task-related conversations, and it achieves intent accuracy of 97.7% and slot F1 of 96.8% [5]. Compared with the rule-based keyword extraction, the pretrained BERT is a context-dependent model that can distinguish the word's meaning in the given context. Therefore, operators may give the same command by using different words, e.g., "Hey Max, I need your help" or "Max, can you give me a hand?" when they need assistance without mentioning the exact keywords.

**Dialogue State Tracking**. A manufacturing task can take several steps to complete. Therefore, a VA may require several rounds or turns of conversation to obtain the necessary slots to complete the task. In this module, a JavaScript object notation (JSON) file called dialogue state rule (DSR) is predefined to specify all the slots (the optional slot marked with "optional", otherwise marked as "Yes") for completing each task. Max verifies and updates the state based on the requested slots. The dialogue continues until Max receives all the required slots.

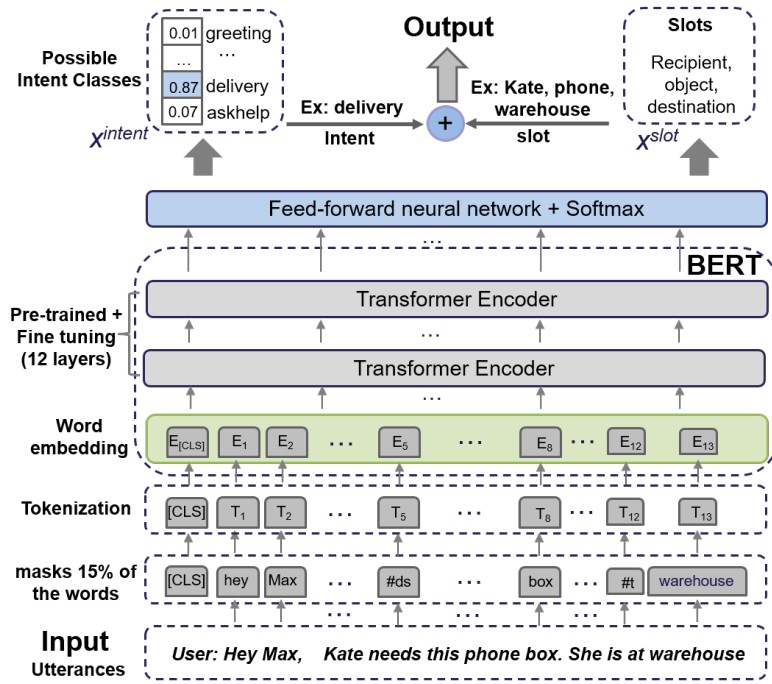

**Figure 3.** The fine-tuned pretrained BERT model for intent prediction and slot extraction.

**Response Generation.** Max's response generation follows the template-based approach, i.e., mapping the nonlinguistic input (e.g., slots values from the user's utterance) to the linguistic surface structure [42]. Furthermore, we introduce human–human conversation strategies in the interaction with the VA and the industrial robots. In this module, two dialogue strategies, namely lexical-semantic strategy and general diversion strategy [43], are integrated so Max can increase the task-completion rate while improving the user engagement. High engagement is also observed from active participation in human–human collaboration, e.g., taking the initiative in a task and smoothly switching the topic if it is not relevant to the current task. In this video (http://y2u.be/lYnh2cOeeE0, accessed on 1 October 2022), we demonstrate Max's ability to initiate a request to the operator and start working with manufacturing tasks at the appropriate time, providing suggestions to the operator if the requested task cannot continue. Therefore, Max can complete the requested tasks and support active dialogues to enhance the user experience.

### 3.2. Robot Service

**Grounding.** Natural language grounding is essential for matching a word to the appropriate representation in terms of terminology and relationship in the learning and training scenarios. At the same time, it is responsible for assigning a vocal command to the associated robot's action representation in the assistance scenario. To support learning scenarios, a knowledge graph (KG) [44], which stores interlinked descriptions of phone production, is designed based on the Neo4j system (https://neo4j.com/, accessed on 1 October 2022). It defines the entities (e.g., materials of producing a phone, operation process), attributes (e.g., color, size), and relationships (e.g., pair, need).

A knowledge graph is defined as

$$KG := (N, E, a_N, r_E, s, t), \tag{1}$$

where

$$N(KG) \text{ is a finite set of nodes (i.e., entities)}$$
$$E(KG) = \{(p, q) \in N \times N\} \text{ where } p, q \in N \text{ is a finite set of edges (i.e., relationship)}$$
$$a_N(KG) : N \to a_N \text{ is a labeling function for nodes attribute}$$
$$r_E(KG) : E \to r_E \text{ is a labeling function for relationship and}$$
$$s(KG), t(KG) : E \to N \text{ attaches the edge to its source and target node.}$$

The grounding process can be explained with the following process. Considering that we have a corpus of operator's utterance $U$ which contains $n$ extracted keywords $K = \{k_1, ..., k_n\}$, we can find a partial function $f$ to map the extracted keywords from operator's utterance to their representations in the graph, $f : K \to KG$. The keywords refer to a predefined set of robot services and skill categories, which are stored as a JSON file in the database. The JSON file also lists the supported robot services and skills for the training and assistance scenarios.

Algorithm 1 provides an example of how the system interprets the operator's dialogue related to the task "delivery_package" to provide the core skill parameters, e.g., "destination" for a package-delivery task.

### 3.3. Voice Interface

A voice-enabled interface is designed for Max's client to enable efficient communication with low hardware requirements (i.e., headphones with a built-in microphone and Bluetooth wireless connection). In this module, we chose the Microsoft cognitive speech services, speech-to-text and text-to-speech, to enable Max's client voice interface. The operator's utterance transcripts are organized into a RESTful style HTTP request and sent to Max's server. The response message is then extracted from the server's reply (a JSON string), and a natural-sounding voice is generated. Such a voice interface frees the operator's hands while providing natural communication for efficient HRI on the shop floor.

---

**Algorithm 1** Robot skill and response computation.

---

1: Input: the dialogue history, $User_0$, $SysAct_0$,...,$User_i$, and DSR
2: Output: the $Robot\_Skill_i$, the $Robot\_Skill\_Parameter_i$ and $res\_ponse_i$
3: extract the current user intent, $intent_i$, and previous intent, $intent_{i-1}$, from $User_i$ and $User_{i-1}$ respectively
4: **if** $intent_i \neq intent_{i-1}$ **then**
5:　　Assign *null* to the $Robot\_Skill_i$ and *inconsistent_intent* to $res\_reference_i$
6:　　return
7: **else**
8:　　Assign $intent_i$ to the $Robot\_Skill_i$
9:　　**for** Every dialogue $D \in (User_0, ..., User_{i-1})$ **do**
10:　　　　save the obtained slots of each turn to *obt_slots*
11:　　**end for**
12: **end if**
13: extract the slots, $slots_i$, from current user utterance, $User_i$
14: **for** Every $slot \in slots_i$ **do**
15:　　**if** $slot \in obt\_slots$ **then**
16:　　　　update the *obt_slots* with the new value of the *slot*
17:　　**else**
18:　　　　save the *slot* to *obt_slots*
19:　　**end if**
20: **end for**
21: extract all the required slots (*req_slots*) from DSR regarding the requested $intent_i$ from $User_i$
22: **for** Every $slot \in req\_slots$ **do**
23:　　**if** $slot \notin obt\_slots$ **then**
24:　　　　Assign *requested_slot* to the $Robot\_Skill\_Parameter_i$ and $res\_reference_i$
25:　　　　return
26:　　**end if**
27: **end for**

---

### 3.4. Web Interface

An interactive web interface is being developed to enhance the operator's experience with LTA-FIT through Max. The interface is designed as a web interface on any web browser-enabled device, such as a mobile phone, tablet, or PC.

Figure 4 illustrates the main web interface. The interface's message panel shows the operator's real-time dialogue with Max. It enables the operator to keep track of the conversation history and verify that the Max accurately transcribes spoken orders and whether the Max delivers relevant responses or not. A robot services panel is included to tell the operator about the services that the requested robot supports. Additionally, a system status panel provides real-time information regarding activities or systems, such as network connections and the status of running processes.

### 3.5. Robot Control Module

Max is designed to be agnostic of robot hardware and aims to support various kinds of industrial robots, from mobile robots to industrial manipulators. Two subcomponents, a robot service management (RSM) component and a robot service execution (RSE) component, are implemented on the robot control module. RSM periodically synchronizes the local robot services and skills with Max's server to update the robot services/skills registered on the local client side and the robot-controlling scripts. RSE invokes the corresponding robot control script, which specifies robot control functionalities (e.g., package delivery) and communication protocols for shop floor robots. A JSON file is maintained on RSM to ground the operator's verbal commands (including operator's intent and slots) to the requested robot service/skill. For example, "warehouse" is grounded to the coordinate

"*x*":, "*y*":, "*z*" of the location of the mobile robot in the 2D map of the warehouse. The grounding results are used to match the corresponding robot control function and set up the input parameters.

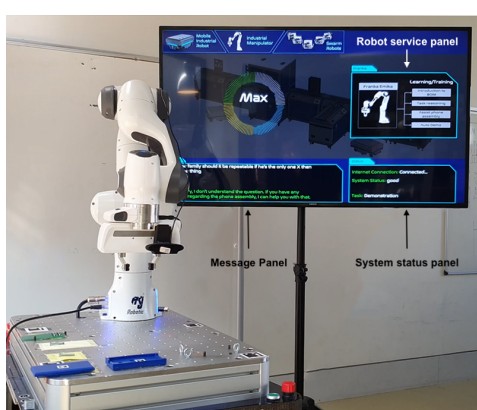

**Figure 4.** The web interface of the Max client running on a common web browser.

## 4. Selected LTA-FIT Scenarios

In order to explore the capabilities of our proposed language-enabled VA, we examined three LTA-FIT scenarios. Here, our focus is to teach the production terminology to the operators, train them to assemble mock-up mobile phones, and assist them with the internal logistics.

### 4.1. Learning Scenario

According to the definition of the LTA-FIT model, learning focuses on providing the appropriate basic knowledge to the workers via instructions. Therefore, in the first scenario, the VA is leveraged to teach the worker the production terminology and the process of the assembly of a mock-up mobile phone (shown in Figure 5).

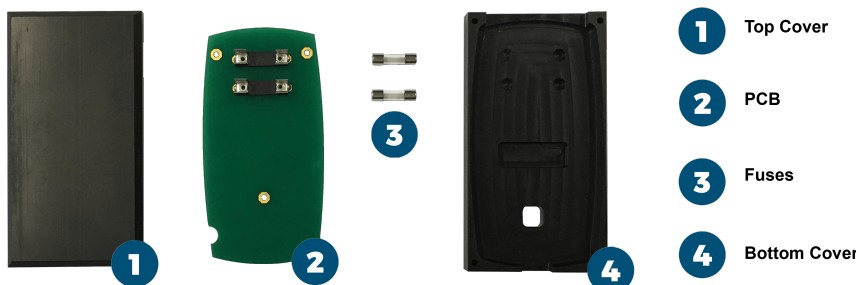

**Figure 5.** The parts of the mock-up mobile phone used for the learning and training scenarios.

Such knowledge consists of a list of the raw materials, components, the respective quantities required to produce a phone, and the corresponding sequence of the phone-assembly task. Usually, such information is shared at the beginning of the training of new operators or during the introduction of new products in production.

In this case, the VA is expected to perform as a question answering (Q&A) system. To this end, we designed a repository of logical connections between the parts of the mock-up phone assembly with the means of the knowledge graph. The knowledge graph contains the core entities (e.g., printed circuit board (PCB), phone cover, fuse), attributes (e.g., the color of the cover), and the relationships (e.g., the order of the operations) regarding the mock-up phone assembly task (see Figure 6 [5]). The VA is able to understand the operator's verbal requests (e.g., "how many processes are needed in a phone assembly?"), retrieve the relevant answer according to the knowledge graph, and generate responses to the operator (e.g., "There are eight operations in total.").

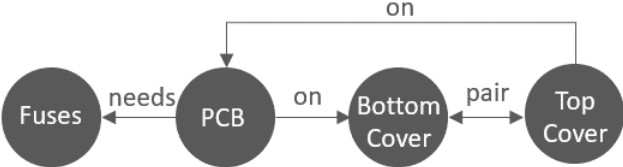

**Figure 6.** The graphic representation of the knowledge graph for the assembly of the mock-up mobile phone.

### 4.2. Training Scenario

To solidify the gained knowledge from the learning level, the second scenario is selected for training operators to assemble the mock-up mobile phone. A live demonstration performed by an experienced and skilled operator is the most common way to carry out such a task in current manufacturing environment with a follow-up Q&A or hands-on learning session afterward. In this scenario, the VA performs as a task-oriented dialogue system. Based on the dialogue with the operator, the VA controls the industrial mobile manipulator and demonstrates the required steps for the assembly of the mock-up phone. The robot manipulator is programmed to perform the mock-up phone assembly automatically. However, the continuous assembly process may be challenging for operators to follow, and the correct sequence of the operations could be difficult to remember. Therefore, in addition to the entire demonstration, the VA is able to guide the operators throughout all the steps of the hands-on learning session.

### 4.3. Assistance Scenario

The third scenario targets the top level of the LTA-FIT, i.e., assistance. The purpose is to evaluate the performance of the VA when it comes to assisting the operators' daily work and, specifically, internal logistics. In general, internal logistics focus on internal supply processes, transportation of materials and tools, and cargo distribution. Such tasks may require a mobile robot to deliver the goods to a location within the organization. A fleet manager or web application for robot control usually comes as a package with the chosen mobile robot to assist in scheduling and tracking the tasks. However, it could have a steep learning curve for new or inexperienced operators. To this end, we tested the VA on two practical assistance cases, including package delivery and system status checking.

The VA is expected to obtain the core information (e.g., a package delivery task may require the destination of the delivery and a recipient) through a dialogue with the operator. In such a case, a standard wireless headset is the only extra device we need to maintain the communication between the operator and VA. The operator's verbal command is transmitted through the headset to the VA. Based on the needs of these two cases, real-time assistance in manufacturing tasks requires that the VA can understand the operator's utterance with high accuracy and low latency. Therefore, in contrast to the learning and training scenarios, the assistance scenario focus on tracking the VA's overall performance based on four quantitative metrics, i.e., intent error rate (IER), slot error rate (SER), task success rate (TSR), and average communication time (ACT).

## 5. Experimental Setup

The industrial feasibility of the three LTA-FIT scenarios was evaluated based on a set of experiments designed to assess the interactions of the human operators with the robot and the VA. The participants were expected to complete a task during the experiments while interacting with the system. The tasks for the learning and training scenarios were focused on the assembly of the mock-up mobile phone (shown in Figure 5), whereas the assistance scenario focuses on an internal logistic task. The list of identified tasks is presented in Table 1).

**Table 1.** Identified tasks for the three LTA-FIT scenarios.

| Category | Task Description | Intent Recognition |
|---|---|---|
| Learning | 1: Explain terminology | Keywords |
| Learning | 2: Reason operation relationships | Keywords |
| Training | 3: Demonstrate auto phone assembly | Keywords |
| Training | 4: Guide phone assembly | Keywords |
| Assistance | 5: Assist package delivery | BERT Model |
| Assistance | 6: Check system status | BERT Model |

The purpose of our experiments was twofold: (1) exploration of the system's behavior when used by a diverse user group; (2) collection of suggestions for improvement on the usability, workload, and overall human–machine interaction. Therefore, we were interested in the following research question: How can the adoption of a natural language-enable virtual assistant for LTA-FIT scenarios provide an easier implementation of Industry 4.0 principles in factories?

*5.1. Evaluation Metrics*

To assess and analyze the interaction between the VA, the robot, and the human operators, subjective and objective metrics were used for the examined scenarios. All participants used a postexperiment questionnaire to rate their interaction with the system anonymously in the learning and training scenarios. This questionnaire was built with standardized questions to evaluate the system's usability, the perceived mental load while performing the task with the system and subjective metrics which cover a comfortability scale, the perceived competence, the performance, and effort expectancies. As mentioned in Section 4.3, four quantitative metrics were used to track the performance of the VA in the assistance scenario. These four metrics can assess Max's ability to recognize the operators' verbal commands, perform tasks successfully, and latency during the interaction with the users.

5.1.1. Metrics Used in the Learning and Training Scenarios

**System Usability Scale.** The SUS is a ten-item survey where the participants score the experiment on a five-point Likert scale (from strongly disagree to strongly agree). Afterward, an overall score of the system can be used to assess its usability of the system. The SUS score is calculated by following the instructions in [45], where each item is attributed a score from 0 (strongly disagree) to 4 (strongly agree). Therefore, a participant's score can vary within the range of 0–100 and in our experiments, and we used the grouping proposed by [35] as: "80–100, users liked using the system"; "60–79, users accept the system"; "0–59, the users did not like to use the system".

**NASA-TLX questionnaire.** The NASA-TLX questionnaire is commonly used to assess the mental workload that users experience while interacting with a system. It includes six subscales that the participants rate individually, i.e., mental demand, physical demand, temporal demand, performance, effort, and frustration [41,46].

Furthermore, the NASA-TLX questionnaire often employs a 20-point or 100-point scale for participants to grade the experiment. However, because there are many areas for the participant to score, there is a possibility for additional frustration while completing the questionnaire. Therefore, several researchers [47,48] support the idea that having a five-point format in the scale design could reduce the frustration level of the users and could thereby increase the response rate. Consequently, we used a five-point Likert scale format in our questionnaire to increase the quality of the responses.

Similarly, we divided the last subscale (frustration) into categories since the original question contains numerous adjectives that might be deemed synonyms. The new cate-

gories we used are: stressed, insecure, and annoyed. These are evaluated in relation to both the interaction with the VA and the robot system overall.

**Subjective metrics.** In addition to the objective and standardized SUS and NASA-TLX scales, we deemed it necessary to utilize some of the subjective metrics proposed by [35] and add our own comfortability scale. Similar to SUS, the participants score using a five-point Likert scale in a postexperiment questionnaire. These subjective metrics cover the following elements:

- The comfortability scale measures the participant's level of comfort with the system at the beginning and conclusion of the interaction.
- Perceived competence refers to the participant's conviction that they can complete the task with the help of the system.
- Performance expectancy is the degree to which participants feel that utilizing the system would help them perform better in a work environment.
- Effort expectancy defines the degree of easiness involved with using the system.

### 5.1.2. Metrics Used in the Assistance Scenario

**Intent Error Rate (IER).** This is the rate where the model cannot predict the requested intent of the user correctly. For instance, the requested intent is PACKAGE_DELIVERY, but the predicted intent from the BERT model might be POSITIONCHECK. IER is obtained by calculating the proportion of it occurring in all experiments.

**Slot Error Rate (SER).** This is the rate of incorrect slot values, i.e., the BERT model selects an incorrect value for the slot, and slot detection fails, i.e., the model cannot recognize the slot. Similar to IER, the number of occurrences of failing to recognize slot values is used to calculate the SER.

**Task Success Rate (TSR).** This corresponds to Max's ability to retrieve all the required information from operator's verbal commands to complete the task without encountering any intent or slot error problem.

**Average Communication Time (ACT).** This is the required time for a task to be completed on average. The average communication time equals the meantime from the beginning of the conversation to its end. In general, high IER or SER requires more communication time.

### 5.2. Physical Setup

The experiments were conducted in our learning factory at Aalborg University [22]. Figure 7 provides a schematic representation of the physical setup, together with a visual representation of the placement of the phone components on the robot table. A camera and an external microphone were placed 2.2 and 1.6 meters away from the robot table, respectively, to capture the participants' interactions for further analysis.

### 5.3. Evaluation Procedure

At the beginning of the experiment, the users were introduced to the overall goal and purpose of the user studies. A short introduction to the scope of the survey was given, along with a brief explanation of the task at hand. Afterward, the users were introduced to the methods they could choose to interact with the system, and they were informed that the whole experiment would be recorded for archival purposes. To conduct the experiments, participants were asked to randomly choose a task and initiate a dialogue based on sample dialogues that were distributed to them. A sample task specification and the corresponding dialogue example of the phone assembly used for the training scenario are shown in Figure 8a, where the user requests from the system to demonstrate how to assemble the phone. In Figure 8b, the VA asks the user to verify that all components are in their default places before starting the demonstration.

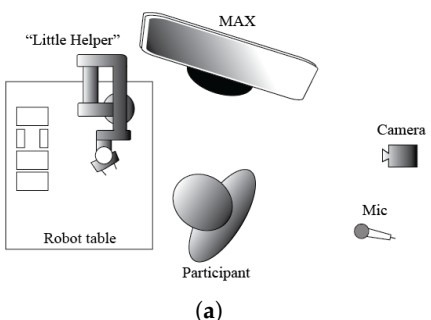

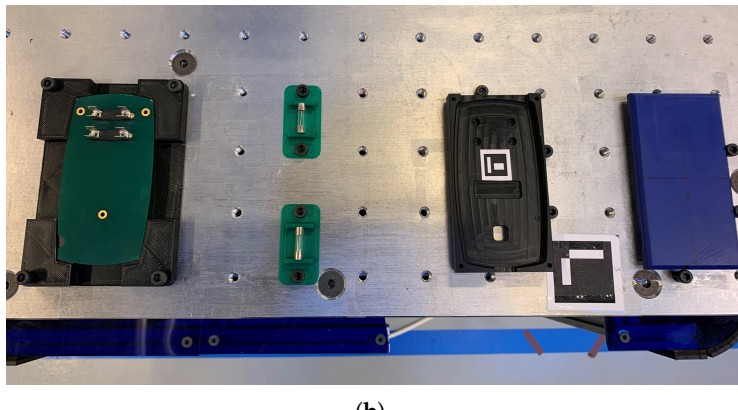

(b)

**Figure 7.** An analysis of our experimental setup. (**a**) Schematic representation of the use case setup. Each user was positioned 30 cm away from the robot platform. The camera and microphone were positioned 1.5 m away and the screen where Max was showing messages for the interaction was placed 80 cm away from the user. (**b**) Phone components on the robot table.

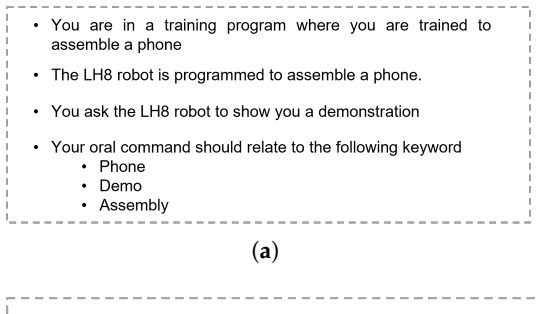

(a)

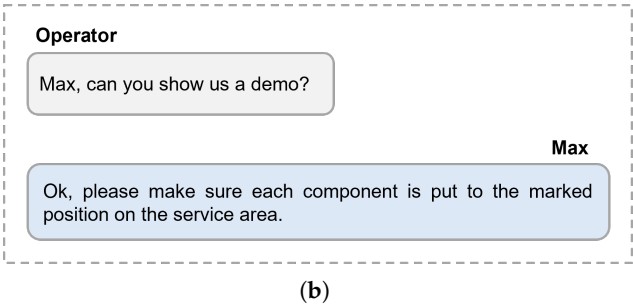

(b)

**Figure 8.** An example of a task and a dialogue for the training scenario. (**a**) A sample task specification for the assembly of the mock-up phone. (**b**) Example dialogue for the task of phone assembly.

In total, 29 people took part in the user study. Based on their demographics shown in Figure 9, they represent the age group of 20–25 years old primarily, they are mainly users who identify as males (72.4%), and 75.9% have an IT/engineering background. Moreover, 72.4% of all users had interacted with an industrial robot before, whereas almost 21% had previous interactions with a VA.

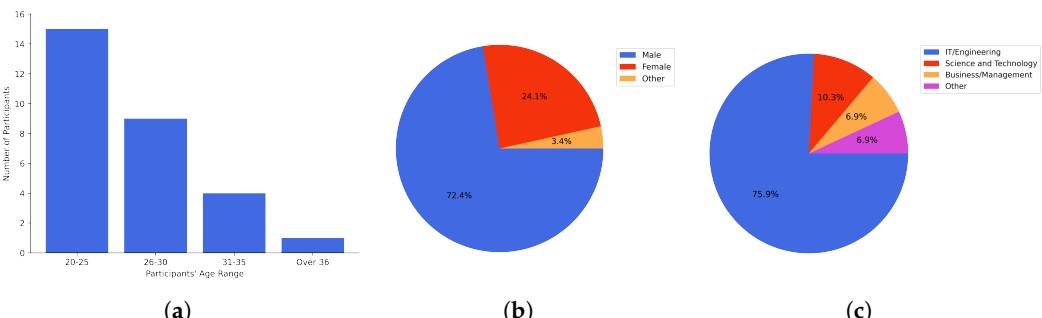

**Figure 9.** Participants' demographics. (**a**) Participants' age range. (**b**) Participants' gender. (**c**) Participants' study/work background.

## 6. Experimental Results

### 6.1. System Usability Scale

In general terms, the users found the system easy to use, and they were satisfied with the interaction with the VA and the industrial manipulator. In our experiments, the SUS score varied between 50.0 and 90.0. The average score in the system usability scale was 68.53, which, as we mentioned in Section 5.1, means that the users accept the system. Obviously, they did not have an enthusiastic response to the system for multiple reasons that we explain in Section 7.

### 6.2. NASA-TLX

In terms of the mental workload that the users had to overcome while using the system, the obtained results with NASA-TLX showcase that the system requires low mental demand (MD) while the users can achieve high performance with minimal effort. Figure 10 depicts the obtained scores for each subscale of the NASA-TLX questionnaire.

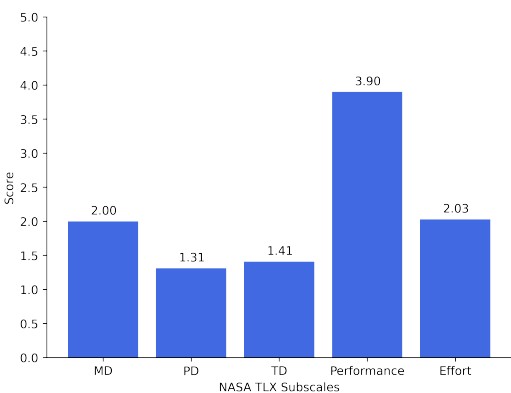

**Figure 10.** The obtained results based on the NASA-TLX questionnaire.

As expected, physical demand (PD) scored the lowest with 1.31 because no physical interaction with the industrial manipulator was needed to complete the required tasks. At the same time, temporal demand (TP) also scored low with 1.41 because there were no time limits or time requirements for the completion of the tasks. Furthermore, it is important to mention that these scores are related to the evaluation of the overall system. However, the overall frustration subscale is divided into three categories, i.e., stressed, insecure, and annoyed, and evaluated for Max and the industrial manipulator separately. In the box-plot shown in Figure 11, we can see that the overall frustration is low for both systems. However, the annoyed category scores higher than expected for the VA. The reason for this reaction from the users were the network delays and lack of quick feedback from the VA when multiple queries took place during the experiments.

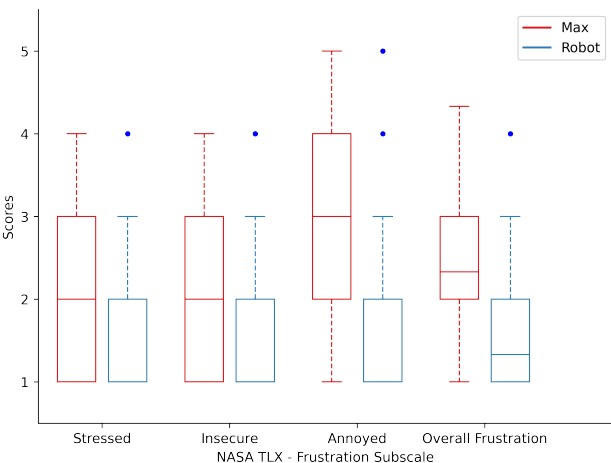

**Figure 11.** Scores obtained for the overall frustration subscale of NASA-TLX.

*6.3. Subjective Metrics*

In a similar fashion, as in the SUS and NASA-TLX metrics, the scores collected for the subjective metrics proved that the system was intuitive to use. More specifically, Figure 12 visualizes the mean score for each metric, where higher scores mean better evaluation of the system. Based on the scores obtained in the comfortability scale (CS), we can safely state that the users found the system comfortable to use, whereas by looking at the scores of perceived competence (PC), they were confident in their competencies to complete the tasks with the help of the system. At the same time, the high score of effort expectancy (EE) proves that the general impression of the users was that the system is easy to use; however, based on the score of performance expectancy (PE), there were still users who were not convinced that the proposed system would assist them adequately in the work environment.

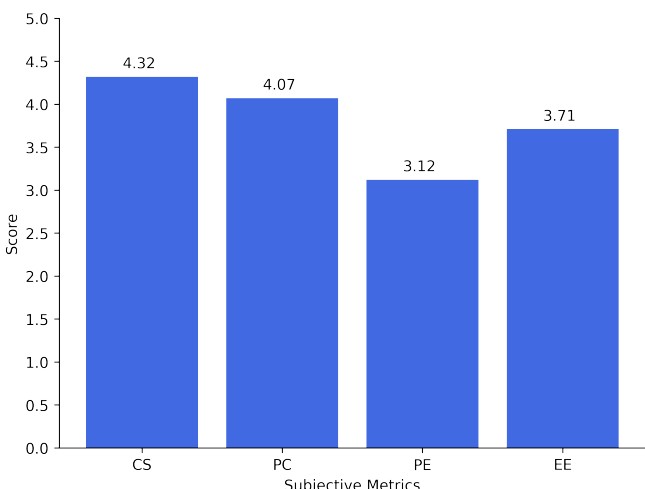

**Figure 12.** Scores obtained for the subjective metrics: comfortability scale (CS), perceived competence (PC), performance expectancy (PE), and effort expectancy (EE).

*6.4. Quantitative Performance Results*

In order to assess the individual performance of the VA in the tasks of the assistance scenario (see Table 1), we performed each task 30 times. The evaluation was divided considering two different areas of interest; the language area and the task area. In the former area, we assessed how well Max can recognize the users' intent and slot value during the dialogue, whereas in the latter we measured the rate where the VA could execute the requested tasks successfully and the average communication costs. As Table 2

shows, the VA managed to recognize the operator's intent with minimum IER but struggled with recognizing the required slot values. Due to the complexity of the dialogues and the ambient noise of an industrial shop floor, the VA completed the two tasks with TSR of 0.50 and 0.76, respectively, whereas it required 31.08 and 15.28 s on average to complete the communication.

**Table 2.** Quantitative results for the performance of the proposed VA in the tasks of the assistance scenario.

| Task ID | IER | SER | TSR | ACT (in secs) |
| --- | --- | --- | --- | --- |
| 5 | 0.1 | 0.43 | 0.50 | 31.08 |
| 6 | 0.2 | 0.06 | 0.76 | 15.28 |

## 7. Discussion & Conclusions

In this work, we demonstrated a natural language-enabled VA, integrated with a robot platform to conduct multiple learning, training, and assistance scenarios in a real industrial environment. In an expansion of our prior work [5], we conducted user studies with an emphasis on user feedback on satisfaction and usability of our system.

The user study proved that adopting the proposed VA could provide an easier implementation of Industry 4.0 technologies in factories and support a more intuitive interaction with intelligent systems and robots. In particular, the system scores high in usability, user satisfaction, and comfortability metrics whereas it requires low cognitive effort from the vast majority of the users. At the same time, the VA alone performs with a high success rate and handles complex queries with minimum error rates.

Naturally, there are several lessons learned and areas for improvement. As many users in the user studies originated from a diverse geographic range, they spoke with various accents and dialects during their interaction with the system. This fact created an unexpected challenge for the VA to understand the verbal commands of certain participants. As a side effect, the VA would occasionally perform an incorrect command or would request the participant to repeat the instruction for clarification multiple times leading to higher annoyance and mental demands, as shown in Figure 11. However, toward the conclusion of the experiments, the participants appeared to have an easier time interacting with the system due to their increased familiarity with the system's behavior. Additionally, most participants expressed satisfaction with the system, with many of them stating that it was "fun and intuitive" and that they would be interested in working with such a system in their workplace.

A significant area for improvement in our future work is the management of ambient noise. Further investigation into noise-suppression methods of steady-state background noise (e.g., ventilation, production) in the workplace can significantly improve the speech recognition accuracy, resulting in a higher task success rate and shorter communication times. Moreover, the addition of speech-enhancement techniques can also be essential to alleviate issues with various dialects and decrease the accent recognition error rate.

Despite the fact that the proposed user study confirmed the satisfaction and usability of the VA, the population of sampled users is limited, and the majority of them are lab engineers, researchers, and students with a robotics background. Future plans include organizing a large-scale user study with our industry partners. Together with the shop floor experts, we will specify their LTA tasks and collect and annotate a new dialogue corpus for training the BERT model in order to implement and assess the VA in the company. Additionally, the robot control algorithms will be adjusted based on the robots in use. Employees and shop floor specialists will be invited to evaluate the VA's generality.

The source code of the proposed VA and the supplementary material of the user study are accessible at https://github.com/lcroy/Virtual-Assistant-Max, accessed on 1 October 2022) to support the reproducibility of the proposed framework.

**Author Contributions:** Conceptualization, O.M., C.L. and D.C.; methodology, C.L. and D.C.; algorithm, C.L. and D.P.; experiments, D.P. and C.L.; research, C.L. and D.C.; writing—original draft preparation, C.L., D.C. and D.P.; writing—review and editing, C.L., D.C., D.P., A.K.H., S.B. and O.M.; supervision, D.C. and C.L. All authors have read and agreed to the published version of the manuscript.

**Funding:** The authors would like to acknowledge support by EU's SMART EUREKA programme under grant agreement S0218-chARmER, Innovation Fund Denmark (Grant no. 9118-00001B), and the H2020-WIDESPREAD project no. 857061 "Networking for Research and Development of Human Interactive and Sensitive Robotics Taking Advantage of Additive Manufacturing—R2P2". The research was also funded by the European Regional Development Fund and University College of Northern Denmark.

**Institutional Review Board Statement:** This experiments were reviewed and approved by the Ethical Committee of Faculty of the Engineering and Science, Department of Materials and Production, Aalborg University.

**Informed Consent Statement:** All participants signed an informed consent form to take part in this study.

**Data Availability Statement:** Not Applicable.

**Acknowledgments:** Authors wish to express their gratitude to all groups of users who graciously volunteered to be part of the testing phase of the prototype.

**Conflicts of Interest:** The authors declare no conflict of interest.

## Abbreviations

The following abbreviations are used in this manuscript:

| | |
|---|---|
| ACT | Average Communication Time |
| AI | Artificial intelligence |
| BERT | Bidirectional Encoder Representations from Transformers |
| CS | Client-Server |
| CS | Comfortability Scale |
| DSR | Dialogue State Rule |
| EE | Effort Expectancy |
| FFNN | Feed-Forward Neural Network |
| HRC | Human-Robot Collaboration |
| HRI | Human-Robot Interaction |
| IER | Intent Error Rate |
| JSON | JavaScript Object Notation |
| KG | Knowledge Graph |
| LH8 | Little Helper 8 |
| LTA-FIT | Learning, Training, Assistant-Formats, Issues and Tools |
| MD | Mental Demand |
| PC | Perceived Competence |
| PCB | Printed Circuit Board |
| PD | Physical Demand |
| PE | Performance Expectancy |
| Q&A | Question Answering |
| RSE | Robot Service Execution |
| RSM | Robot Service Management |
| SER | Slot Error Rate |
| SUS | System Usability Scale |
| TLX | Task Load Index |
| TP | Temporal Demand |
| TSR | Task Success Rate |
| VA | Virtual Assistant |

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
