# Peer review of "Hey Max, Can You Help Me? An Intuitive Virtual Assistant for Industrial Robots"

_applsci, doi:10.3390/app13010205_

Round 1

Reviewer 1 Report

It shows 85% similarity to a published article by the same author(s). Please consult with the editor. 

Author Response

We would like to thank the reviewer for the comments. We have already consulted by the editor who accepted the manuscript.  

Reviewer 2 Report

Dear Authors

This is an interesting article about the Virtual Assistant for Humans collaborating with Industrial Robots, therefore the title is a bit confusing.

The article is well prepared, but some aspects require clarification and supplementation.

Description of the robot platform is missing.

The used robot "Little Helper" is described in the previous work [5], but for the convenience of the readers, a brief description of the components, including Franka Emika Panda cobot and the MiR 200 mobile robot, is recommended.

The problem of human safety during collaboration with robots is completly omitted.

In this work the human-robot interaction is achieved through spoken natural language, which should be safe for humans as there is a cobot with safety mechanism already built in used.

In addition, most users have an IT / engineering background and have had previous contact with an industrial robot and are already familiar with safety requirements and hazards.

In the case of new users, there should be a OSH instruction before beginning the work with the robot. Because of risk of improper human behavior or reasonably foreseeable misuse. 

Minor comments

A graphic representation of a knowledge graph defined in line 190 would be helpful for the readers

 The abbreviation of JSON is not described

 Other comments

The problem of relations between humans and robots is broader.

Most discussions on robot development draw on the Three Laws of Robotics devised by Isaac Asimov:

·         -robots may not injure humans (or through inaction allow them to come to harm);

·         -robots must obey human orders (even if they are wrong);

·         -and robots must protect their own existence.

Therefore, the question is:

What should a robot do when a human is giving him wrong orders?

Reviewer 3 Report

This article presents a Virtual Assistant, based on natural language processing architecture, to assist humans in complete assembly tasks in HRI.

The methodology is well described and supported by an experimental survey, concerning a mobile phone assembly task.

There are some minor issues, which are described as follows. I would recommend the authors to make revisions accordingly in order to improve this manuscript. Before considering this paper ready for publication, minor revisions are mandatory.

- Figure 2 is barely legible; I suggest to increase the font if possible, introduce a legende and use different colors for different boxes;

- Section 3.2: the description is very articulate and unclear. Revise the writing to make it more fluent;

- Figure 5a: please, add the distances between objects in order to make the experimental scenario clearer;

- Row 394: 29 participants seems to be a low number for this type of investigation. Do you think it is sufficient?

- The experiment should always be in accordance with ethical requirements. I'm sure it is, but it needs to be clarified in the article. Has the experiment been approved by an Ethical Committee or similar?

Reviewer 4 Report

The paper develops an intuitive virtual assistant called Max for engieering robots in Industry 4.0 background. 29 users were used to test the framework and the usability and cognitive efforts were investigated. The design is valid and relevent to real practice. Therefore, it would be useful to the society. Just a few minor suggestions for improvememt:

1. The thoery and fundamental principles of the systems need further clarification by using equations and diagrams.

2. It is suggested to further discuss the applicability and limitations of the current developement in various ocassions, and give comments on how the proposed framework can be volumely deployed.

3. Explain why these metrics used are sufficient in evaluating the performances. 
